# Freezing and thawing magnetic droplet solitons

Martina Ahlberg [1,8], Sunjae Chung [1,2,8✉], Sheng Jiang[1,3,4], Andreas Frisk [1], Maha Khademi[5], Roman Khymyn [1], Ahmad A. Awad [1], Q. Tuan Le[1,4], Hamid Mazraati[4,6], Majid Mohseni[4,5], Markus Weigand [7], Iuliia Bykova[7], Felix Groß [7], Eberhard Goering[7], Gisela Schütz[7], Joachim Gräfe [7] & Johan Åkerman [1,4✉]

Magnetic droplets are non-topological magnetodynamical solitons displaying a wide range of complex dynamic phenomena with potential for microwave signal generation. Bubbles, on the other hand, are internally static cylindrical magnetic domains, stabilized by external fields and magnetostatic interactions. In its original theory, the droplet was described as an imminently collapsing bubble stabilized by spin transfer torque and, in its zero-frequency limit, as equivalent to a bubble. Without nanoscale lateral confinement, pinning, or an external applied field, such a nanobubble is unstable, and should collapse. Here, we show that we can freeze dynamic droplets into static nanobubbles by decreasing the magnetic field. While the bubble has virtually the same resistance as the droplet, all signs of low-frequency microwave noise disappear. The transition is fully reversible and the bubble can be thawed back into a droplet if the magnetic field is increased under current. Whereas the droplet collapses without a sustaining current, the bubble is highly stable and remains intact for days without external drive. Electrical measurements are complemented by direct observation using scanning transmission x-ray microscopy, which corroborates the analysis and confirms that the bubble is stabilized by pinning.

[1] Department of Physics, University of Gothenburg, 412 96 Gothenburg, Sweden. [2] Department of Physics Education, Korea National University of Education, Cheongju 28173, Korea. [3] School of Microelectronics, Northwestern Polytechnical University, 710072 Xi'an, China. [4] Department of Applied Physics, School of Engineering Sciences, KTH Royal Institute of Technology, 100 44 Stockholm, Sweden. [5] Department of Physics, Shahid Beheshti University, Evin, 1983969411 Tehran, Iran. [6] NanOsc AB, 164 40 Kista, Sweden. [7] Max Planck Institute for Intelligent Systems, Stuttgart, Germany. [8] These authors contributed equally: Martina Ahlberg, Sunjae Chung. ✉email: sjchung76@knue.ac.kr; johan.akerman@physics.gu.se

Magnetic droplets are intrinsically dynamic, non-topological, magnetodynamical solitons[1–15], which can be nucleated and sustained both in spin-torque nano-oscillators (STNOs)[3,6,8,12,14] and spin Hall nano-oscillators (SHNOs)[16], provided the magnetodynamically active layer has sufficient perpendicular magnetic anisotropy (PMA). Magnetic

droplets are characterized by a reversed core separated from the surrounding magnetization via a perimeter of precessing spins (see Fig. 1a)[2,3]. While first predicted over 40 years ago in an ideal zero-damping medium[1], their possible experimental realization was later suggested theoretically[2] in STNOs with PMA-free layers[17,18]. After the first experimental demonstration of

**Fig. 1 Droplet vs. bubble, device structure and layout, and magnetic characterization. a** Schematic of dynamical magnetic droplet soliton. **b** Schematic of a static magnetic bubble. **c** Schematic of an all-perpendicular STNO composed of [Co/Pd] (fixed) and [Co/Ni] (free) multilayers with a Cu spacer fabricated on a SiN membrane structure. The narrow area in the middle of the mesa is designed to easily locate the NC. The insets underneath show optical micrographs of the SiN membrane areas through which the different metal layers of the device can be seen. **d** Hysteresis loops of single Co/Pd and Co/Ni layers. **e** Hysteresis loop of a full [Co/Pd]/Cu/[Co/Ni] stack.

magnetic droplets, reported in STNOs with a PMA Co/Ni free layer and a Co fixed layer[3], interest in magnetic droplets continues to increase due to their interesting characteristics, such as a highly nonlinear dynamics[2,11,19], large power emission[3,10,20,21], and possible applications in microwave-assisted magnetic recording (MAMR)[22,23] and neuromorphic chips as nonlinear oscillators[24–26]. Several theoretical[5,11,15,19,27–33] and experimental[6–10,12,16,21,34–38] studies on magnetic droplets have since been presented.

As pointed out by Hoefer et al., the droplet is reminiscent of a magnetic bubble[2] (Fig. 1b) and they identify a possible zero-frequency droplet with a topologically trivial magnetic bubble[39–43]. Despite a large number of experimental droplet studies, the low-field/low-frequency behavior of droplets has not yet been explored and the relation between droplets and bubbles —as well as a possible transition between the two—remain unclear.

In this work, in order to explore these phenomena, we have studied magnetic droplets specifically in the low-field regime using both electrical and microwave spectroscopy measurements as well as direct microscopical observations based on Scanning Transmission X-ray Microscopy (STXM). We found clear experimental evidence for a droplet-to-bubble transition as the field strength, and hence the droplet frequency, was reduced, and a reversible bubble-to-droplet transition as the field was again increased in an attempt to squash the bubble, provided stabilizing spin-transfer torque was still present via the STNO current. Our experimental results hence corroborate the picture, first expressed by Hoefer et al., that a magnetic droplet can be viewed "*as an imminently collapsing bubble that is critically stabilized by the localized injection of spin torque*".

## Results and discussion

**Samples**. Figure 1 shows a schematic of the studied all-perpendicular STNOs, comprised of a [Co/Pd]/ Cu/[Co/Ni] giant magnetoresistance (GMR) stack deposited on a $Si_3N_4$ membrane (for fabrication details, please see Methods). Underneath the schematic we show two optical microscopy images taken from opposite directions to highlight the optical transmission of the $Si_3N_4$ membrane. In Fig. 1d we show the magnetic properties of the individual free and fixed layers based on calibration samples, and their combined behavior in full STNO stacks in Fig. 1e.

**Droplet to bubble transition**. Figure 2 presents the resistance and microwave signal as a function of the field for an applied current of −5 mA. The field is first increased from −0.51 to 0.51 T in Fig. 2a and then decreased from positive to negative field in Fig. 2b. At large negative fields, the STNO is in its lowest resistance state, consistent with a parallel (P) relative orientation of its free and fixed layers. At about −0.49 T, the resistance increases about 20 mOhm in a step-like fashion and there is a slight increase in the microwave noise background, both strong indications of the nucleation of a droplet. The all-perpendicular symmetry makes it impossible to harvest the characteristic frequency of the droplet perimeter since the in-plane precession does not contribute to the magnetoresistive signal. The symmetry can be broken by applying the external field at an angle $\Theta_H < 90°$. Figure S1 in the Supplementary Materials demonstrates the observation of GHz excitations in measurement using $\Theta_H = 30°$. The low-frequency noise originates from drift instabilities[2] (i.e., the droplet escapes the NC) and droplet mode hopping[14]. At about −0.38 T, there is a second step-like increase in the STNO resistance and a marked further increase in the microwave noise. We interpret this as a transition into a larger droplet as the

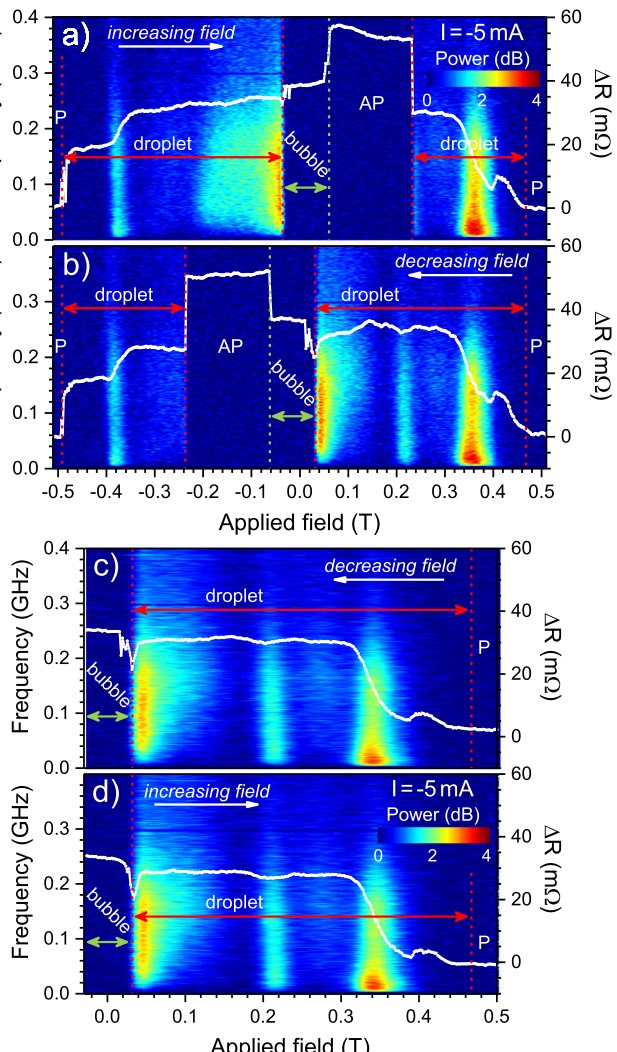

**Fig. 2 Microwave noise and STNO resistance vs. field. a–d** Color plot of the power spectral density (PSD) of the microwave noise as a function of increasing (**a**, **c**) and decreasing (**b**, **d**) field, with the STNO resistance (white line) overlayed; the applied current is −5 mA. **a**, **b** Wide field sweep covering full saturation at both positive and negative fields. P/AP indicate the parallel/antiparallel state of the STNO; red arrow indicates the droplet region, and green arrow the bubble region. **c**, **d** Minor field sweeps showing how the droplet/bubble transition is fully reversible.

opposing applied field is reduced. At yet lower fields the droplet continues to grow in size (the STNO resistance increases), while its stability seems to deteriorate as indicated by the growing intensity of the microwave noise background. At about −0.04 T, the microwave noise rapidly reaches a maximum and then suddenly disappears altogether, while the resistance exhibits a small jump of about 5 mOhm. The complete microwave silence indicates that the magnetic state is now static, and we are led to conclude that the droplet precession has stopped entirely and that the droplet has transitioned into a nanobubble state.

The nanobubble resistance exhibits a jump reminiscent of Barkhausen noise[44–46], indicating pinning possibly at grain boundaries or defects of the sputtered film. When the field is further increased, the bubble resistance increases gradually, indicating a continued growth of its size. At about 0.06 T, the entire free layer switches its magnetization direction and the antiparallel (AP) state is clearly identifiable in the resistance. This increase in resistance is caused by the switching of the entire free

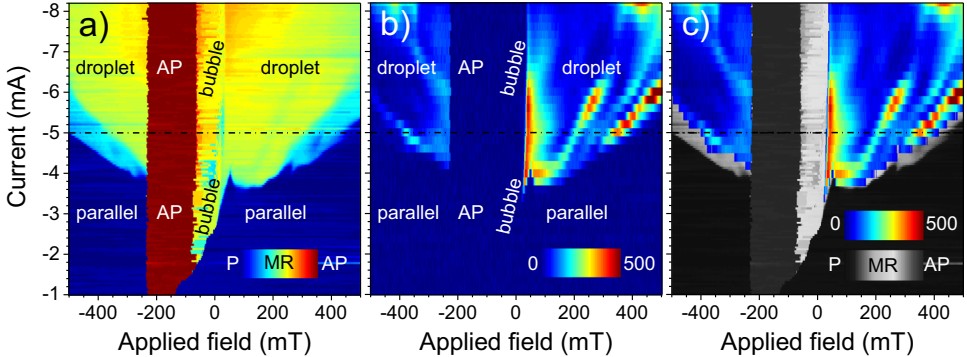

**Fig. 3 Phase diagrams based on the resistance and the microwave noise. a** STNO resistance and **b** integrated (0–0.5 GHz) microwave noise level as a function of field and current. Panel **c** shows the noise level in **b** overlaid on the resistance in **a** displayed using a grayscale highlighting intermediate resistance levels indicative of droplets/bubbles. The dash-dotted black line corresponds to the field sweep at $I = -5$ mA given in Fig. 2. The parallel (P) and antiparallel (AP) states are easily discernible in the MR-map (**a**) as dark blue and dark red, while both the droplet and the bubble are characterized by intermediate resistance in green–yellow. The stark difference between the droplet and the bubble is revealed in the noise spectrum (**b**), where the stability of the bubble is manifested. Note, however, that the light-blue flanges in **a** correspond to a different droplet regime not captured in the microwave signal presented in **b**.

layer throughout the device (Fig. 1c), which enhances the magnetoresistance compared to the bubble state where only a small volume below the NC is reversed. When the fixed layer switches at 0.23 T, a droplet is immediately nucleated. With further increase of the opposing field, the droplet again shows a gradual transition to a smaller size; the droplet finally disappears as the STNO transitions into a full P state at about 0.47 T. The overall behavior is very similar for decreasing fields (Fig. 2b) where the same P/AP/droplet/nanobubble states can be clearly identified via the STNO resistance and the microwave noise.

As mentioned above, the microwave noise power is far from constant for the whole droplet region and peaks at certain fields. We identify these peaks as marks of mode hopping between different droplet states[14]. While the details of the spectrum are highly reproducible (cf. increasing and decreasing fields) and serve as a fingerprint for each device, the patterns at negative and positive fields are quite different. The magnetoresistance implies that a relatively small and stable droplet ($\mu_0 H < -0.4$ T) is abruptly followed by a larger but similarly stable mode. In contrast to the symmetric noise patterns around the droplet-to-droplet transitions, the strong increase in microwave noise power around the droplet-to-bubble transition is highly asymmetric. There is first an extended field region of monotonic increase in the noise, which is then abruptly cut off and replaced by a completely silent bubble state. This highlights the very different non-dynamical nature of the nanobubble and suggests that mode hopping out of the nanobubble state and back into a droplet state is negligible, once the nanobubble has formed.

**Freezing and thawing**. Figure 2c, d demonstrates that it is possible to freeze the dynamic droplet into a static bubble and then thaw it back into a droplet using only the magnetic field under constant spin-transfer torque. In particular, Fig. 2d shows how the nanobubble first is about to collapse at 0.025 T as it is getting squeezed by the opposing pressure from the increasing applied field. There is some slight Barkhausen-like noise in the rapidly dropping resistance, but otherwise no measurable microwave noise. However, instead of switching to a P state, the resistance then exhibits a sharp minimum after which it shows a rapid increase, which is accompanied by a high level of microwave noise. The collapsing nanobubble is hence rescued by the stabilizing spin-transfer torque, which sets the spins in the bubble perimeter into precessional motion and restores the full dynamics of a magnetic droplet. Judging from the resistance, it is

noteworthy that the droplet is slightly larger than the smallest nanobubble. Within the experimental accuracy (a field step of 2 mT), we do not observe any significant hysteresis in this transition. Hence there is a negligible energy barrier between the two states and the bubble can indeed be viewed as a zero-frequency droplet, albeit still likely affected by pinning.

**Phase diagram**. Figure 3a presents a phase diagram based on a two-dimensional map of the STNO resistance as functions of current and field. All data were acquired in a decreasing field at a constant current level. The parallel (P) and antiparallel (AP) configurations are easily identified by the dark blue and dark red colors, respectively, and for current magnitudes below 1.8 mA, these are the only two available states, as expected for a GMR device. However, even at these weak currents, the P → AP switching field is clearly affected by the STT from the nano-contact; in contrast, the AP → P switching field is entirely unaffected. In an intermediate current region, from about −1.8 to −3.5 mA, the STT can not yet sustain a droplet but is sufficient to create a nanobubble directly from the P state. As magnetic switching typically involves both domain nucleation and domain propagation, we interpret this current dependent switching in the following way (see Supplementary Materials, Fig. S2, for a zoom-in of this particular part of the phase diagram). For current magnitudes below 1 mA, magnetic switching is limited by the field required for domain nucleation and, in addition, the location of initial domain nucleation is far from the nanocontact region as STT from the current has no discernible impact. However, for current magnitudes above 1 mA, where we observe a strong current dependence of the switching field, we conclude that the domain nucleation has moved underneath the nanocontact. If we reduce the field magnitude, we need a stronger current to assist in the domain nucleation, but once formed, it propagates through the entire free layer. However, at fields weaker than the field needed for domain propagation, i.e., the pinning field, which we read out as about 60 mT, the nucleated domain is no longer able to propagate and instead remains as a nanobubble directly underneath the nanocontact. The nanobubble can hence form either from the P state or from a droplet. Once formed, the bubble is stable even without a sustaining current, see Fig. S3 in Supplementary Material.

The droplet shows two discernable states, a high-field/low-current mode that exhibits a rather small MR (light blue). This mode moves to higher fields with increasing current and is no

longer visible above ≈ −6 mA. The other distinguishable droplet mode is characterized by an intermediate resistance (green–yellow). The bubble is almost indiscernible from the latter droplet state, even though a subtle line traces the transition between the two. Moreover, the bubble resistance is not a smooth function of the applied field, but displays notches and steps, indicative of Barkhausen-like noise due to pinning.

In contrast to their almost identical resistance, a stark difference between the droplet and the bubble is uncovered in Fig. 3b, where we show the microwave signal integrated over 0–0.5 GHz. The droplet exhibits non-zero power levels of low-frequency microwave noise, while the P, AP, and bubble states are definitely static and silent. Figure 3b also further unveils the complex relationship between the applied field and current, and the particular droplet characteristics. A strong microwave noise signal denotes mode hopping[14] and these events exhibit a strong dependence on both field and current. We can identify three traces of mode hopping for positive fields, while there is only two weak trails at negative fields. There are also regions where the droplet is very stable and the noise level is almost zero. These features act like fingerprints for each measuring device and are highly reproducible in consecutive measurements, but differ between STNOs. We then overlay the microwave noise data onto the resistance data, now plotted with a grayscale that highlights intermediate resistance levels (Fig. 3c). Parts of the low-field/low-current droplet regime (light blue in Fig. 3a) do not exhibit any measurable microwave noise. It is possible that its dynamics is on a slower time scale than the microwave frequencies our setup is sensitive to.

**STXM imaging and simulations**. We now turn to the results of the scanning transmission X-ray microscopy results, illustrated in Fig. 4. Images of the droplet/bubble are shown in Fig. 4a–f, and the corresponding magnetoresistance and microwave signal are presented in Fig. 4g with the matching field of the images marked by their letter. The STXM and the electrical measurements were performed in separate setups, hence there is a small uncertainty in comparing the field values of the two, although the images clearly correspond to the magnetic states expected from the electrical signal. The dashed white or black circles mark the position of the nanocontact. It has been placed by assuming that the droplet/bubble in Fig. 4d is centered under the NC and by comparing the non-magnetic contrast of the different images. The non-magnetic contrast used were the white and black spots on the left side and in the middle of the NC, respectively, visible in the inset of Fig. 4g. The method works very well as confirmed by the good overlap of the perimeters in the inset of Fig. 4g, but it should be remembered that the absolute position is still based on this assumption.

Figure 4a is measured at 270 mT, and shows a mode associated with a high noise level in Fig. 4g. Only a weak and mostly white feature is captured in the STXM image. STXM measures a time-averaged image and the droplet is in this highly noisy regime expected to experience large drift instabilities and continuously vanish and renucleate underneath the nanocontact. As a consequence, only a washed out and poorly reversed feature results. In contrast, Fig. 4b–d display more stable and more clearly reversed droplets. They have approximately the same radius as the nanocontact, although the size definitely increases slightly with decreasing field, as expected. We have in earlier STXM work observed a significant effect of the Zhang-Li torque on the droplet size[21,47]. The magnitude of this effect depends on the current density ($j_{dc}$) and we have performed simulations that confirm that the difference between the droplet diameter here and in our former publication is indeed due to a weaker $j_{dc}$. At zero field, a bubble is

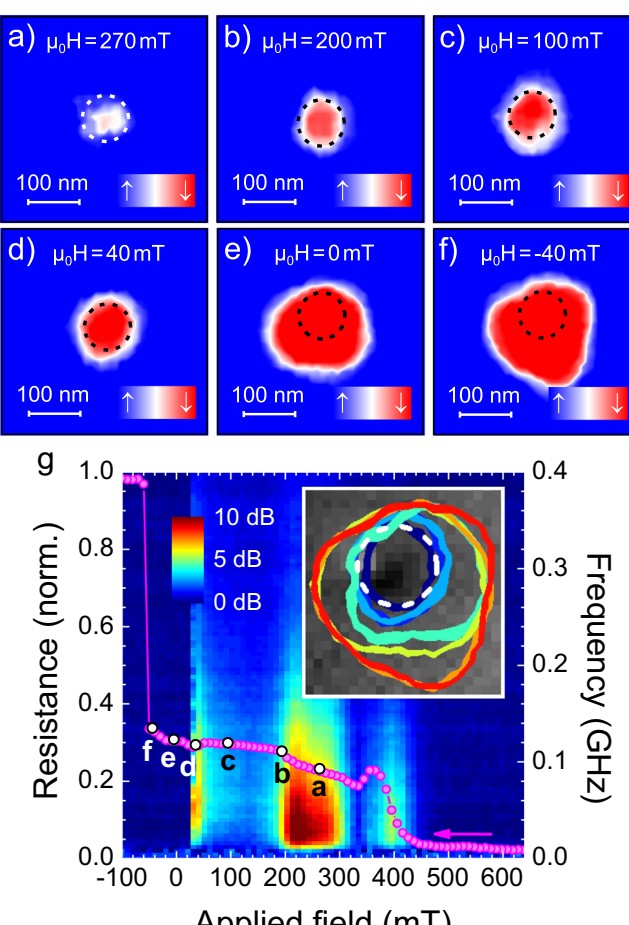

**Fig. 4 Scanning transmission X-ray microscopy (STXM). a–f** STXM images of the nanocontact region vs. decreasing field for a current of −7 mA. Blue corresponds to magnetization aligned with the applied field, red corresponds to magnetization anti-aligned with the applied field, whereas white indicates zero out-of-plane magnetization. The STNO resistance and the microwave noise PSD vs. decreasing field are shown in **g** where the points corresponding to the STXM images have been labelled **a–f**. The inset in **g** highlights the perimeter of the droplet/bubble as the applied field is decreased from 200 mT (dark blue) to 60 mT (blue), and further reduced to −40 mT (red) in steps of 20 mT.

clearly formed and it prevails down to −40 mT (Fig. 4e, f). It is no longer centered on the nanocontact but has mostly expanded in one direction. It should be noted though, that the field in the microscope is given by rotating permanent magnets and the sample may have been subjected to in-plane fields between two set values. Nevertheless, the inset in Fig. 4g presents the perimeter of the droplet/bubble as the field decreases from 200 mT (dark blue) to −40 mT (red), and the initial bubble at 40 mT (light blue) grows in distinct steps, which implies that the size is controlled by pinning.

We finally carry out micromagnetic simulations to corroborate the experimental results on the freezing and thawing of droplets and bubbles. A key ingredient in these simulations is the inclusion of pinning. A wide range of different random distributions of key magnetic material parameters was investigated to mimic the pinning properties of a polycrystalline film (see Methods). Our simulations confirm the importance of the pinning landscape and, depending on the choice of material parameter distributions, the droplet can freeze into a static bubble at positive, zero, or negative fields. Once frozen, the bubble can always be thawed back to a droplet by increasing the field, and an example of such

freezing/thawing of a droplet is given in Fig. S4 in the Supplementary Materials, which illustrates the evolution of the magnetization dynamics together with snapshots of the magnetic states at different fields. The simulation results clearly corroborate the distinct experimental observations of (i) a (noisy) dynamic droplet transforming into a static bubble, stabilized by pinning, (ii) a Barkhausen-like growth of the bubble with the reduced or negative field, (iii) stability of the bubble when the field and current are removed, and (iv) thawing of the bubble back to a droplet once the field is again increased at constant current.

Returning to the original droplet theory of Hoefer et al., we note that pinning was not included[2]. It is clear from our experimental observations and simulations that pinning has a strong and immediate impact on the relation between droplets and nanobubbles and must be included in the low-field/low-current regime. Instead of exhibiting a continuous slow-down and frequency decrease to zero with decreasing field, there is a minimum droplet precession frequency that spin-transfer torque can sustain before pinning overcomes the precession. As this minimum frequency is approached from above, the broad-band microwave noise diverges as the competition between the inertia of the precession and the pinning makes the droplet dynamics increasingly erratic until pinning finally gets complete control of the precession abruptly stops, leaving complete microwave silence in its wake.

## Methods

**Sample preparation**. The sample stack consisted of a Ta (4 nm)/ Cu (14 nm) / Ta (4 nm) / Pd (2 nm) seed layer and an all-perpendicular pseudo-spin valve [Co (0.35 nm) / Pd (0.7 nm)] × 5 / Co (0.35 nm) / Cu (5 nm) / [Co (0.22 nm) / Ni (0.68 nm)] × 4 / Co (0.22 nm), capped by a Cu (2 nm) / Pd (2 nm) layer, which was deposited by magnetron sputtering on a Si wafer with 300 nm-thick LPCVD silicon nitride layer. Using conventional photolithography and metal-etching techniques, 8 × 16 μm mesas were fabricated on the stacked wafer and all mesas were insulated by a 30 nm-thick SiO₂ layer deposited using chemical vapor deposition (CVD). Electron beam lithography was used to pattern nanocontacts (NCs) on the top of each mesa having different diameters from 50 to 150 nm. The SiO₂ layer was then etched through by the reactive ion etching (RIE) technique to open NCs. The NC-STO device fabrication was completed by deposition of a Cu (200 nm) / Au (100 nm) top electrode and lift-off processing. For STXM measurements, Si was removed from the backside using a highly selective RIE process leaving only the SiN membrane to allow X-ray transmission underneath the NC-STOs (see Fig. 1c). For magnetic and electrical characterization of the NC-STOs, the same stack was prepared on a thermally oxidized Si wafer and then a similar fabrication processing was done, except for the deep etching for a membrane structure.

**Magnetic and electrical characterization**. The magnetization hysteresis loops were measured using Alternating Gradient Magnetometry (AGM) with the unpatterned material stacks. Microwave and dc measurements of the fabricated STOs were carried out using our custom-built setup, where magnetic field strength, polarity, and angle can be controlled. A magnetic field between −0.5 to +0.5 T can be manipulated using an electromagnet. The device is connected using GSG probe to a dc-current source (Keithley 6221), a nanovoltmeter (Keithley 2182A), and a spectrum analyzer (R & S FSQ26). A 0–40 GHz bias-tee is used to separate the bias input and the generated microwave signal. The microwave signal is amplified by a low-noise amplifier (operational range: 0.1–26.5 GHz) before being sent to the spectrum analyzer. The data presented in Figs. 2, 3, as well as in Figs. S1, S2, S3 in the Supplementary Materials, is collected using a device with a NC diameter ($d_{NC}$) of 70 nm, while Fig. 4 display data acquired for a sample with $d_{NC} = 90$ nm.

**Scanning transmission x-ray microscopy**. The STXM measurements were performed at the BESSY II synchrotron, using the MPI IS operated MAXYMUS end station at the UE46-PGM2 beamline[48]. The out-of-plane component of the magnetization was probed using circularly polarized light at normal incidence. The applied field, with a maximum value of 300 mT, was generated by a set of four rotatable permanent magnets[49]. An optimal XMCD contrast was achieved by setting the photon energy to the Ni $L_3$ edge, which resulted in clear images. The size of each pixel is $10 \times 10$ nm², while the nominal resolution of the focusing plate is 18 nm.

**Micromagnetic simulations**. Micromagnetic simulations of the free layer magnetodynamics were performed using the GPU-based finite-difference micromagnetic solver MuMax3[50]. The STNO was modeled by $512 \times 512 \times 1$ cells with a cell size of $3.9063 \times 3.9063 \times 3.9063$ nm³. Absorbing boundary conditions in the form of a smoothly increasing damping profile were applied to the simulated sample edges to avoid any interference artifacts from spin-wave reflection. The uniaxial magnetic anisotropy $K_u = 519$ kJ/m³ was determined by out-of-plane FMR measurements, giving an effective magnetization of $\mu_0 M_{eff} = -0.55$ T, together with the literature value of the saturation magnetization $M_s = 716.2$ kA/m[5]. The exchange stiffness was set to $A_{ex} = 10$ pJ/m and the damping constant was $\alpha = 0.03$. The charge current distribution and the resulting Oersted field landscape was calculated using COMSOL Multiphysics (www.comsol.com) using experimentally determined layer resistivities: [Ta/Cu/Ta](22 nm) 9 μΩ × cm; Pd/[Co/Pd] (7.6 nm) 22.1 μΩ × cm; Cu(4 nm) 4.74 μΩ × cm; [Co/Ni]/Co(3.8 nm) 27.9 μΩ × cm; [Cu/Pd](4nm) 8.1 μΩ × cm. The calculated current profile and the Oersted field were then supplied to MuMax3, together with the following parameters: current polarization $P = 0.4$ and spin-torque asymmetry parameter $\Lambda = 1.3$, and degree of non-adiabaticity $\xi = 0.013$[21]. Pinning points were introduced by dividing the simulation area into a random distribution of regions representing grains using the Voronoi tessellation extension to MuMax3[51]. To simulate grain boundaries in the film, the intergrain exchange coupling was scaled randomly using a uniform distribution in different ranges, for the figure shown the range from 40 to 60% of the bulk value was used. In addition, for each grain $K_u$ was set to random values ± 10% of the nominal value. We observed that the simulations are very sensitive to the random pinning landscape. Furthermore, we noticed that for these parameters set at random the choice of ranges of values is less important than the shape and the distribution of the grains, which are determined by the random seed. Further tests also included random values of $M_s$ or in-plane cubic anisotropy for each grain. Multiple combinations of parameter variations were tested, of which many gave droplet to bubble transitions, but the most reliable approach to create pinning points was to apply random intergrain exchange coupling. The droplet was nucleated in an applied field and current, whereafter the current was kept constant while the field was decreased to a value where the droplet froze into a bubble, consecutively the field was increased again thawing the bubble to a droplet. Most simulations were performed with $T = 0$ K, but tests were also run at $T = 300$ K for completeness, which did not show any qualitatively different behavior.

## Data availability

The data used to produce the plots within this paper are available at figshare.com [https://doi.org/10.6084/m9.figshare.19493789]. All other data used in this study are available are available from the corresponding author on reasonable request.

## Code availability

The MuMax3 codes generated and analyzed during the current study are available at figshare.com [https://doi.org/10.6084/m9.figshare.19493789].

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

## Acknowledgements
This work was supported by the Swedish Research Council (VR; 2017-06711 and 2019-04229) (J.Å.). Helmholtz Zentrum Berlin is acknowledged for allocating beam time at the BESSY II synchrotron radiation facility. M.W., E.G., G.S., and J.G. acknowledge the financial support by the Federal Ministry of Education and Research of Germany in the frame work of DynaMAX (Project No. 05K18EYA). This work was supported by the National Research Foundation of Korea(NRF) grant funded by the Korea government (MSIT) (No. 2020R1F1A1049642) (S.C.).

## Author contributions
M.A., S.C., and J.Å. conceived the project, S.C., S.J., and T.Q.L. performed the electrical measurements. S.C., T.Q.L., S.J., and A.H. fabricated the devices. M.A., S.J., J.G., M.W., F.G., and I.B. carried out the STXM measurements. A.F. and M.K. carried out the micromagnetic simulations with assistance from A.A.A. and M.M.J.Å. coordinated the project. All authors analyzed and discussed all results and co-wrote the manuscript.

## Funding

## Competing interests
The authors declare no competing interests.
