## [Peer Review File · Nature Communications]

Reviewers' Comments:

Reviewer #1:

Remarks to the Author:

I thoroughly enjoyed reading the manuscript submitted by Ahlberg et al. It is very well written and in a manner that it can be easily understood and followed "in one run". In the manuscript the authors describe how they generate droplets in a magnetic multilayer and convert these droplets into magnetic bubbles and vice versa. As correctly described by the authors these items seem very similar at first but are distinguished by being either a static or dynamic object. For this reason they chose an approach in which they evaluate the occurrence and power of microwave noise in dependence of current through the device and applied external field. States without noise are static and states with noise are dynamic, indicating the presence of a droplet. The static(noise less) states can be attributed to either the P or the AP state of the multilayer. The occurrence of a third static state strongly indicates the presence of a bubble. This assumption is then corroborated at the end using x-ray microscopy. The authors then show that they can transform the system from a droplet to a bubble state and vice versa.

The findings are of significant importance to the community for reasons laid out by the authors in their introduction. The methods are sound and the conclusions are well supported. Overall I find that the manuscript certainly warrants publication in Nature Comm. and will most likely have significant impact in the community and beyond. For this reason I recommend publication with minor revisions as listed below:

Page 10: Out of curiosity. The authors mention that the different noise regions act as a fingerprint for each device, which makes a lot of sense. Is it possible to elaborate a bit more on this issue, e.g. if certain shapes, thicknesses, defects can affect the noise distribution. This is a minor point that would be of interest to the reader, but it is possible that there is no room to address this appropriately.

Page 10: The statement "seem highly consistent" is a bit weak. it would be better to give an observable reason why they are comparable.

Page 10: Does this mean that the authors can see the NC or not in the chemical images? This was not clear to me.

Page 12: The authors state that the "white color" represents in plane magnetization. But I am not sure if they can really resolve or claim this in particular for images b-f. In most images the white border is smaller than the optical resolution, meaning the border between bubble and background could be sharp, but simply does not get resolved. It could also be possible that these areas are not static but oscillate (like for a droplet). To be sure that this is static in plane contrasting needs to rotate the sample and acquire XMCD images in other geometries.

Reviewer #2:

Remarks to the Author:

This manuscript reports an experimental study of spin-torque—driven magnetization oscillations in perpendicularly-magnetized multilayers. The focus is on dynamical solitons called droplets, which represent an excitation involving large-angle magnetization precession that results from the competition between viscous damping and spin-torques. A particular feature studied is the transition from droplet states to bubble states, where the latter can be stable in the absence of drive currents and resemble regions of reversed magnetization. The authors combined electrical noise spectroscopy with scanning transmission electron microscopy measurements to characterize the range of existence of the droplet excitations.

The subject of magnetic droplet excitations has been explored in some depth, both experimentally and theoretically, for about a decade. The present work represents some incremental progress toward understanding transitions between droplets, which are dynamical solitons (i.e., states that only exist in the presence of current-driven torques) and bubble states, which can be topological

solitons (i.e., states that are metastable under zero applied current). However, I find the evidence rather circumstantial and unconvincing. While there is no doubt that some form of current-driven excitation resembling a droplet is excited, the experiments do not provide anything more conclusive beyond this observation. There are number of serious deficiencies in the manuscript, which are discussed below.

First, the power spectral density of the excitations exhibit microwave noise over a large range of frequencies, as shown in Figure 2, rather than well-defined peaks as discussed in the seminal work by Mohseni et al (Science 2013). Moreover, there does not appear to be any quantitative argument as to how droplets and bubbles produce the power spectra shown in Fig. 2.

Second, the authors suggest that the bubble regime is accompanied by Barkhausen noise, although little evidence is provided to support this observation. Have different field sweeps been performed at low fields to illustrate jumps? Do the authors have time domain data that illustrate random, thermally-activated hopping between pinning sites? The authors also state that energy barriers are negligible in their system, which appears to contradict their observation of Barkhausen jumps.

Third, I am not sure if the term "freezing" is appropriate here. If the transition under study really involves dynamical droplets and bubble states, then perhaps the problem should be cast in terms of nucleation and annihilation. Are the authors suggesting that the bubble state is necessarily non-topological, and that no change in topology is expected between the two states?

Fourth, the authors make references to mode hopping in Figure 3, but no time domain data is given. What do they mean by mode hopping this context? Do they mean a transition from one type of mode to another? This is not very clear.

Fifth, I would contend that bubble states should be (meta)stable in the absence of drive currents, yet the phase diagram in Figure 3 seems to contradict this. How can the authors be sure that there is a real distinction between these two states?

Finally, I feel that comparison with micromagnetics simulations would have been helpful to shed light on the transition between the droplet and bubble states, assuming that latter is in fact observed in this system. As it stands, the so-called "freezing" transition observed remains speculative.

Overall, the manuscript reflects a solid experimental study on magnetic droplet excitations, but it falls glaringly short of the kind of level expected for Nature Communications in terms of depth and scope. I suggest therefore that the authors submit their work to a more specialized journal such as Physical Review B.

Reviewer #3:
None

GÖTEBORGS UNIVERSITET INSTITUTIONEN FÖR FYSIK

We thank the reviewers for their comments and questions. We agree with them and feel that we have now addressed all their comments in an appropriate manner and, as a result, our manuscript has greatly improved. In particular, we have carried out a wide range of micromagnetic simulations to complement our experimental results. This amounted to substantial additional work, where four new coworkers and collaborators contributed. They were also part of the process of revising the manuscript and are now added to the author list. Below we list all questions and our answers and also describe the exact changes that we have made to the manuscript.

Reviewer 1

I thoroughly enjoyed reading the manuscript submitted by Ahlberg et al. It is very well written and in a manner that it can be easily understood and followed "in one run". In the manuscript the authors describe how they generate droplets in a magnetic multilayer and convert these droplets into magnetic bubbles and vice versa. As correctly described by the authors these items seem very similar at first but are distinguished by being either a static or dynamic object. For this reason they chose an approach in which they evaluate the occurrence and power of microwave noise in dependence of current through the device and applied external field. States without noise are static and states with noise are dynamic, indicating the presence of a droplet. The static(noise less) states can be attributed to either the P or the AP state of the multilayer. The occurrence of a third static state strongly indicates the presence of a bubble. This assumption is then corroborated at the end using x-ray microscopy. The authors then show that they can transform the system from a droplet to a bubble state and vice versa.

The findings are of significant importance to the community for reasons laid out by the authors in their introduction. The methods are sound and the conclusions are well supported. Overall I find that the manuscript certainly warrants publication in Nature Comm. and will most likely have significant impact in the community and beyond. For this reason I recommend publication with minor revisions as listed below:

Q1: *Page 10: Out of curiosity. The authors mention that the different noise regions act as a fingerprint for each device, which makes a lot of sense. Is it possible to elaborate a bit more on this issue, e.g. if certain shapes, thicknesses, defects can affect the noise distribution. This is a minor point that would be of interest to the reader, but it is possible that there is no room to address this appropriately.*

A1: It would indeed be interesting to explore the origins of the details in the noise spectra. Unfortunately, at present we do not have enough data to elaborate on this issue. In the literature [Statuto19], multiple droplet modes and observations of transitions between them, including the microwave noise generated in this process, have been found to be device-dependent and highly reproducible within one given device, which is similar to the fingerprinting ideas that we provide. In that reference, a more thorough analysis of this particular aspect is provided.

Q2: *Page 10: The statement "seem highly consistent" is a bit weak. It would be better to give an observable reason why they are comparable.*

A2: We agree and have rephrased that statement. It now reads: "The STXM and the electrical measurements were performed in separate setups, hence there is a small uncertainty in comparing the field values of the two, *although the images clearly correspond to the magnetic states expected from the electrical signal.*"

GÖTEBORGS UNIVERSITET
INSTITUTIONEN FÖR FYSIK

Q3: Page 10: Does this mean that the authors can see the NC or not in the chemical images? This was not clear to me.

A3: We cannot discern the NC per se, but we do observe spots close to the NC and we can use these spots to track the NC position from image to image. We have added the following sentence (italic) to clarify this:

“The dashed white or black circles mark the position of the nanocontact. It has been placed by assuming that the droplet/bubble in Fig. 4(d) is centered under the NC and by comparing the non-magnetic contrast of the different images. *The non-magnetic contrast used were the white and black spots at the left side and in the middle of the NC, respectively, visible in the inset of Fig. 4(g).* The method works very well as confirmed by the good overlap of the perimeters in the inset of Fig. 4(g), but it should be remembered that the absolute position is still based on this assumption.”

The weak chemical signature of the NC makes it hard to find at the start of the microscope measurements (one needs large magnification and magnetic contrast). We have therefore designed the device to reduce the search area and thereby facilitate the locating of the NC. We have added a sentence to the caption of Fig. 1 about the design: “*The narrow area in the middle of the mesa is designed to easily locate the NC.*”

Q4: Page 12: The authors state that the “white color” represents in plane magnetization. But I am not sure if they can really resolve or claim this in particular for images b-f. In most images the white border is smaller than the optical resolution, meaning the border between bubble and background could be sharp, but simply does not get resolved. It could also be possible that these areas are not static but oscillate (like for a droplet). To be sure that this is static in plane contrasting needs to rotate the sample and acquire XMCD images in other geometries.

A4: The referee correctly points out that we could have been more specific with the terminology. We cannot resolve any in-plane components due to the experimental geometry and the observation could also be a net, time-averaged, effect. Therefore, we have changed the caption to “*whereas white indicates zero out-of-plane magnetization.*”

We can, however, resolve the (time-averaged) gradual transition (the domain wall in the bubble case and the precessing region in the droplet case) between state up and state down and it is larger than the spatial resolution. In addition, we do not base our conclusions on the DW being static on the images, but on the electrical signal. We have measured the dynamics using a tilted field, which reveals in-plane oscillations of the droplet and also confirms that the bubble is static. We have added Fig. S2 with this measurement to the Supplementary Material.

Reviewer 2

GÖTEBORGS UNIVERSITET INSTITUTIONEN FÖR FYSIK

This manuscript reports an experimental study of spin-torque—driven magnetization oscillations in perpendicularly-magnetized multilayers. The focus is on dynamical solitons called droplets, which represent an excitation involving large-angle magnetization precession that results from the competition between viscous damping and spin-torques. A particular feature studied is the transition from droplet states to bubble states, where the latter can be stable in the absence of drive currents and resemble regions of reversed magnetization. The authors combined electrical noise spectroscopy with scanning transmission electron microscopy measurements to characterize the range of existence of the droplet excitations.

The subject of magnetic droplet excitations has been explored in some depth, both experimentally and theoretically, for about a decade. The present work represents some incremental progress toward understanding transitions between droplets, which are dynamical solitons (i.e., states that only exist in the presence of current-driven torques) and bubble states, which can be topological solitons (i.e., states that are metastable under zero applied current). However, I find the evidence rather circumstantial and unconvincing. While there is no doubt that some form of current-driven excitation resembling a droplet is excited, the experiments do not provide anything more conclusive beyond this observation. There are number of serious deficiencies in the manuscript, which are discussed below.

Q1: *First, the power spectral density of the excitations exhibit microwave noise over a large range of frequencies, as shown in Figure 2, rather than well-defined peaks as discussed in the seminal work by Mohseni et al (Science 2013). Moreover, there does not appear to be any quantitative argument as to how droplets and bubbles produce the power spectra shown in Fig. 2.*

A1: We appreciate that the reviewer made us aware that the observed PSD needs a bit more elaborate discussion. One needs to remember the differences in sample layout and material stack between this study and [Mohseni13] (as well as in the literature at large). We have a symmetric, all-perpendicular, geometry, where the anisotropy axis of both the free and the reference layers are aligned with the field. This makes it impossible to harvest electrical signals from in-plane dynamics, such as the fundamental frequency of the droplet perimeter precession. Furthermore, the well-defined sidebands in Mohseni *et al.* (and other studies) are caused by well-defined perimeter modulations [Xiao17]. The noise we observe is different in origin and arise from drift instabilities caused by symmetry breaking conditions, which here are mainly represented by the Oersted-field [Hoefer10]. Similar low frequency noise has also been attributed to droplet mode hopping by Statuto *et al.* [Statuto19].

We have added the following sentences to the main text, and a figure in supplementary material, to clarify this to the readers.

“[...], both strong indications of the nucleation of a droplet. *The all-perpendicular symmetry makes it impossible to harvest the characteristic frequency of the droplet perimeter, since the in-plane precession does not contribute to the magnetoresistive signal. The symmetry can be broken by applying the external field at an angle $\Theta_H < 90^\circ$. Figure S2 demonstrates the observation of GHz excitations in a measurement using $\Theta_H = 30^\circ$. The low frequency noise originates from drift instabilities [Hoefer10] (i.e. the droplet escapes the NC) and droplet mode hopping [Statuto19].*”

Q2: *Second, the authors suggest that the bubble regime is accompanied by Barkhausen noise, although little evidence is provided to support this observation. Have different field sweeps been performed at low fields to illustrate jumps? Do the authors have time domain data that illustrate random, thermally-activated hopping between pinning sites? The authors also state that energy*

GÖTEBORGS UNIVERSITET
INSTITUTIONEN FÖR FYSIK

barriers are negligible in their system, which appears to contradict their observation of Barkhausen jumps.

A2: The reviewer gives good advice on how to further explore the exact nature of the observed jumps in resistance. However, we do not claim that we detect explicit Barkhausen noise, neither are our main conclusions based on the origin of the observed jumps. We only state that the distinct jumps in resistance are reminiscent of Barkhausen noise. We argue that our cautious description is valid and have changed the words “Barkhausen noise” on page 8 and 9 to “Barkhausen-like noise”, to further underline that we do use careful expressions.

We thank the reviewer for highlighting that our statement on a negligible energy barrier needs clarification. There is a difference between energy barriers controlling the size of the bubble, and a potential energy barrier between the droplet and bubble modes. The former exists, is caused by pinning, and manifests itself in resistance jumps (i.e. bubble size jumps). The latter is negligible, as shown by the minute hysteresis between the droplet-to-bubble and bubble-to-droplet transitions.

Q3: Third, I am not sure if the term “freezing” is appropriate here. If the transition under study really involves dynamical droplets and bubble states, then perhaps the problem should be cast in terms of nucleation and annihilation. Are the authors suggesting that the bubble state is necessarily non-topological, and that no change in topology is expected between the two states?

A3: We value the reviewer’s concerns, but disagree and argue that “freezing” is appropriate here. We observe a transition from a *dynamic* droplet to a *static* (frozen) bubble and show that the transition is fully reversible. We believe that the words “nucleation and annihilation” should be reserved for the transitions between uniform states (P, AP) and localized states, regardless if the confined magnetic mode is dynamic or static (droplet or bubble).

We do not state anything about the bubble topology, since we simply do not have the means to measure it. However, the droplet is non-topological and it is rational to assume that the bubble inherits this characteristic.

Q4: Fourth, the authors make references to mode hopping in Figure 3, but no time domain data is given. What do they mean by mode hopping in this context? Do they mean a transition from one type of mode to another? This is not very clear.

A4: Yes, we mean mode hopping between different droplet modes and between the droplet and bubble states. A study by Statuto *et al.* [Statuto19] describes that low-frequency noise is associated with droplet mode hopping, but we missed that citation in the discussion on page 10. We sincerely thank the reviewer for making us aware of this omission and we have updated the manuscript accordingly.

Q5: Fifth, I would contend that bubble states should be (meta)stable in the absence of drive currents, yet the phase diagram in Figure 3 seems to contradict this. How can the authors be sure that there is a real distinction between these two states?

A5: The reviewer’s hypothesis, that the bubble is (meta)stable, is completely correct. The reason why it might seem that Fig. 3 contradicts this, is that all the data is acquired in a decreasing field at a

GÖTEBORGS UNIVERSITET INSTITUTIONEN FÖR FYSIK

constant current level. To show that the bubble is stable without drive, one has to first generate a bubble and then switch of the current. This protocol is used in Fig. S3, which we have added to demonstrate the bubble stability. The distinction between the droplet and bubble states are therefore manifested in both zero-drive robustness and noise level. The droplet collapses without drive and although the droplet is remarkably stable for some fields/currents, the bubble is consistently silent.

We have added the following sentence to the manuscript: “The nanobubble can hence form either from the P state or from a droplet. *Once formed, the bubble is stable even without a sustaining current, see Fig. S3 in Supplementary Material.*”

Q6: *Finally, I feel that comparison with micromagnetics simulations would have been helpful to shed light on the transition between the droplet and bubble states, assuming that latter is in fact observed in this system. As it stands, the so-called “freezing” transition observed remains speculative.*

A6: We fully agree that micromagnetic calculations are a valuable tool to elucidate experimental findings. We have therefore performed such simulations and the results are presented in the main text, and in Fig. S4 in Supplementary Material. The results reveal that the droplet-to-bubble freezing and bubble-to-droplet thawing transitions indeed are reproduced micromagnetically. It is also clear that the pinning landscape plays an important role for the bubble stability.

Reviewer 3

The manuscript with title “Freezing and thawing magnetic droplet solitons” reports a study of magnetic states, both static and dynamic, in a spin-torque oscillator based on a bilayer structure of two perpendicular magnetic layers (one polarizer and one free layer). Authors focus the study to the low field range and monitor the formation of either magnetic droplets (dynamic states) or magnetic bubbles (static states). They found that at low fields bubbles arise whereas as the field increases droplets appear (in a continuous and reversible way). The transition between magnetic states (probably due to the low frequency precession of droplets at low fields) is a new phenomenon that links the two magnetic objects. There were almost no studies focusing on the low field regime (one needs perpendicular polarizing layers to have enough polarization in the electrical current). The scope of the study is clear and so it is the scientific question posed. I believe the findings represent a good step forward in the field of nanomagnetism and should be of interest of many researchers with possible technological implications. I could recommend publication of the manuscript.

Q1: *One small question I have refers to the choice of experimental conditions. As I understand, the transition from droplet to bubble occurs because the precession frequency of droplet is getting close to zero and there are other phenomena dominating the stabilization of the magnetic objects (pinning or simply dipolar fields). Could still droplets exist at zero applied field? Under which conditions? What is the precession frequency of the observed droplets? Or at least, is it known the droplet frequency (I do not mean the low-f noise) at the moment of the transition?*

A1: The reviewer has understood the cause of the transition correctly. No, the droplet is not expected to exist at zero applied field, since it should be stabilized as a static structure for fields larger than zero [Hofer10]. The precession frequency is not measurable and hence not known (see Q1, referee 2).

GÖTEBORGS UNIVERSITET
INSTITUTIONEN FÖR FYSIK

However, we have included a measurement using a tilted applied field in the supplementary material, Fig. S2, which shows a GHz-frequency signal. In a simple model the droplet-to-bubble transition field of 40 mT correspond to a perimeter frequency of 3 GHz. (Using Eq. (2) in Ref. [Xiao17] and $\gamma/2\pi=30.5$ GHz/T (gyromagnetic ratio), $M_s = 716.2$ A/m (saturation magnetization), $K = 519$ kJ/m³ (uniaxial anisotropy), $A_{ex} = 7$ pJ/m (exchange stiffness) and $\rho_0 = 50$ nm (droplet radius).)

Q2: *It is not clear to me why the AP state seems to have a much larger R compared to droplet or even bubble. Figure 4 indicates it should be the same as the area beneath the NC is totally red.*

A2: This is caused by the current path from the NC to the ground pads. Only a small volume under the NC is reversed in the bubble state, while the free and fixed layers still are parallel for the remainder of the current path all the way to underneath the ground pads. The total voltage drop from signal to ground is the sum of the droplet and the remaining current path, and this current path is mostly in-plane from the droplet to ground. In the AP state the entire free layer has switched and the resistance is therefore larger.

We have added the following sentence to the main text: “[...] and the antiparallel (AP) state is clearly identified in the resistance. *This increase in resistance is caused by the switching of the entire free layer throughout the device (Fig. 1c), which enhances the magnetoresistance compared to the bubble state where only a small volume below the NC is reversed.*”

Q3: *Bubbles may also have dynamics, because they have domain walls and domain wall resonances are in the MHz regime. I guess that high-frequency data would prove the existence of droplets or at least be much more conclusive, otherwise it could be that all the observed data corresponds to simply bubbles.*

A3: We have included a new figure (Fig. S2) in the supplementary material that shows a measurement with the external field applied at an angle. The data reveals a GHz-signal, and the low frequency noise can thus not simply be attributed to bubble dynamics. Furthermore, the existence of droplets in our all-perpendicular system is most expected, considering earlier theoretical and experimental studies.

Q4: *In page 2 it is said that “Magnetic droplets are characterized by a reversed core separated from the surrounding magnetization via a perimeter of precessing spins” Does this mean the core is not precessing?*

A4: It depends on how reversed the perimeter is. If it is fully reversed the core is essentially not precessing. Figure 1c) in Ref. [Mohseni13] shows a good illustration of the precessing angle along the droplet border. This is also very nicely discussed and illustrated in the original paper by Hofer et al. [Hofer10].

Q5: *In page 6 it is said “the patterns at negative and positive fields are quite different”. I think this was already observed. Is there a clear explanation? Could it be simply spin torque asymmetry?*

GÖTEBORGS UNIVERSITET
INSTITUTIONEN FÖR FYSIK

A5: No, we do not have a clear explanation to the observed behaviour, but we know that it is not simply due to the spin torque asymmetry. We have tested the effect of the spin torque asymmetry in simulations and it fails to reproduce the data.

Q6: If I understand it well, if a bubble is formed it could remain stable for a long time but as long as the current is switch off it disappears, doesn't it?

A6: No, the bubble is stable in zero current (see Q5, referee 2). We have added Fig. S3 in the Supplementary Material to highlight that the bubble is stable without drive.

Q7: Would one of the conclusions be that it is difficult or almost impossible to go from bubble to droplet by varying applied current? I mean this is a relevant feature regarding technology applications.

A7: Yes, it is difficult or almost impossible to go from bubble to droplet by varying the applied current. We have so far not succeeded to do this experimentally. We have, however, gone from bubble to droplet with current in simulations, but this does not always happen and we have so far not studied this in detail.

References

- Hofer10; Hofer et al., Phys. Rev. B, **82**, 054432 (2010).
Mohseni13; Mohseni et al., Science, **339**, 1295 (2013)
Statuto19; Statuto et al., Phys. Rev. B, **99**, 174436 (2019)
Xiao17; Xiao et al. Phys. Rev. B, **95**, 024106 (2017)

Reviewers' Comments:

Reviewer #1:

Remarks to the Author:

I find the response to my previous comments satisfactory. As I pointed out in my first review I find this manuscript suitable for Nat Comm and this has not changed by the revisions. Therefore I now recommend publication as is,

Reviewer #2:

None

Reviewer #3:

Remarks to the Author:

Authors have replied to all my queries and comments.

I also feel the questions of other reviewers are well addressed so I recommend the publication of the manuscript.